# Substrate binding plasticity revealed by Cryo-EM structures of SLC26A2

Wenxin Hu [1], Alex Song[1] & Hongjin Zheng [1] ✉

SLC26A2 is a vital solute carrier responsible for transporting essential nutritional ions, including sulfate, within the human body. Pathogenic mutations within SLC26A2 give rise to a spectrum of human diseases, ranging from lethal to mild symptoms. The molecular details regarding the versatile substrate-transporter interactions and the impact of pathogenic mutations on SLC26A2 transporter function remain unclear. Here, using cryo-electron microscopy, we determine three high-resolution structures of SLC26A2 in complexes with different substrates. These structures unveil valuable insights, including the distinct features of the homodimer assembly, the dynamic nature of substrate binding, and the potential ramifications of pathogenic mutations. This structural-functional information regarding SLC26A2 will advance our understanding of cellular sulfate transport mechanisms and provide foundations for future therapeutic development against various human diseases.

Sulfate ion ($SO_4^{2-}$) is the fourth most abundant anion in human plasma and plays a crucial role in various biological processes[1]. For example, one of its essential functions is as a key component in cellular growth and development through sulfate conjugation[2]. This conjugation is a crucial step for the biotransformation and detoxication of various foreign substances, known as xenobiotics[3]. In addition, through sulfation, $SO_4^{2-}$ acts as a regulator of the bioactivity of endogenous molecules like steroids, peptide hormones, and neurotransmitters, influencing their functions and interactions[4]. Moreover, $SO_4^{2-}$ is responsible for a post-translational modification called tyrosine sulfation, which occurs in various soluble and membrane proteins[5]. In the human body, the primary source of inorganic $SO_4^{2-}$ stems from the active uptake of water and food, which is facilitated by specific solute carriers (SLCs) within the lipid bilayers. These transporters include sodium-dependent SLC13 and sodium-independent SLC26 members[2]. The active transport of $SO_4^{2-}$ is essential for maintaining the required sulfate levels in various tissues and organs, contributing to the proper functioning of the body's physiological processes.

The SLC26 family consists of eleven multifunctional anion exchangers and channels that transport a broad range of substrates, including chloride ($Cl^-$), bicarbonate ($HCO_3^-$), oxalate ($C_2O_4^{2-}$), iodide ($I^-$), formate ($HCO_2^-$), and $SO_4^{2-}$[6]. SLC26A2 is highly expressed in many tissues during development and postnatally prominently in enterocytes and chondrocytes[7,8]. The significance of SLC26 members is evidenced by their association with genetic disorders and clinical conditions[9]. Pathogenic mutations in SLC26A2 result in a spectrum of phenotypes, ranging from severe and lethal diseases, such as achondrogenesis type 1B and atelosteogenesis type 2, to milder disorders like diastrophic dysplasia and multiple epiphyseal dysplasia[10]. Interestingly, the expression of SLC26A2 seems context-dependent and linked to various diseases. For instance, SLC26A2 is upregulated in Crohn's disease[11]. In contrast, SLC26A2 is downregulated in colon cancer as the repression of SLC26A2 in colon cancer cells in vitro increases proliferation[12]. Moreover, a recent study reported that SLC26A2 acts as an unusual mediator of TRAIL resistance, as the loss of SLC26A2 resensitizes cancer cells to TRAIL-mediated cell death[13]. These findings make SLC26A2 an important target for further investigation in understanding and potentially treating related diseases.

SLC26A2 is known to function as an electroneutral anion exchanger, facilitating the exchange of various substrates such as $SO_4^{2-}$ and $C_2O_4^{2-}$ for $Cl^-$, hydroxide ($OH^-$), or even with each other[14,15]. The transport mode is promiscuous, as it is bidirectional and depends on the $Cl^-$ and $OH^-$ gradient across the membrane[16]. To understand the molecular mechanism of SLC26A2, it is crucial to have high-resolution structural information. However, the available structural information about SLC26 members has primarily focused on human SLC26A5

[1]Department of Biochemistry and Molecular Genetics, University of Colorado Anschutz Medical Campus, School of Medicine, Aurora, US.
✉e-mail: hongjin.zheng@cuanschutz.edu

(prestin) and human SLC26A9, along with their close homologs from other species[17–23].

In this work, using cryo-electron microscopy (cryo-EM) single particle reconstruction, we determine three high-resolution structures of human SLC26A2 in complex with various substrates (Cl⁻, $C_2O_4^{2-}$, $SO_4^{2-}$) at 3–3.6 Å resolution. By combining molecular dynamics simulations, mutagenesis, and transport assays, we illustrate the molecular details of how SLC26A2 interacts with substrates that bound to distinct sites. We also discover a unique feature for the dimerization of SLC26A2, which involves a specific stretch of residues in the N-terminal loop (residues 53–61). This unique feature, distinct from the typical β-sheet arrangement in STAS domains of well-known SLC26 members, adds a different dimension to our understanding of its structural dynamics. Furthermore, we map pathogenic mutations identified in patients onto our high-resolution structures and provide logical explanations regarding their potential impact on the transporter function.

## Results

### The overall structure of SLC26A2

For structural and functional research described in this study, we expressed full-length human SLC26A2 with an N-terminal His-tag in HEK293 cells. We purified the overexpressed SLC26A2 in Lauryl Maltose Neopentyl Glycol (LMNG) detergent with 150 mM NaCl at pH 7.5 and then determined the structure, termed SLC26A2-Cl⁻, by cryo-EM (Fig. 1). The final reconstruction achieved a resolution of ~3.2 Å, with sufficient quality for model building (Supplementary Fig. 1, Supplementary Movie 1). The resulting model covers residues 52–724 without several flexible loops containing residues 186–210, 319–333, and 617–644. Like its homologs[17–24], SLC26A2 forms a domain-swapped homodimer (Fig. 1a). Each protomer of SLC26A2 can be divided into three regions: the N-terminal region (residues 1–104) comprising a few flexible cytosolic helices and loops (Supplementary Movie 2), the transmembrane domain (TMD, residues 105–540) with 14

transmembrane helices (TMs) (Supplementary Movie 3), and the C-terminal cytosolic sulfate transporter and anti-sigma factor antagonist domain (STAS, residues 541–739) (Supplementary Movie 4). The TMD adopts a typical uracil transporter (UraA) fold, where the first seven TMs inversely relate to the last seven TMs with a pseudo two-fold symmetry[25]. Furthermore, these 14 TMs are organized into two distinct sub-domains in TMD: the core domain formed by TMs 1–4, 8–11, and the gate domain formed by TMs 5–7, 12–14 connecting to the STAS domain (Fig. 1b). The substrate translocation pathway is located within the interface between the core and gate domains. The two sub-domains undergo an elevator-like rigid-body movement to facilitate substrate transport, enabling the transporter to switch between inward-facing and outward-facing conformations (Supplementary Fig. 2).

In the SLC26A2 dimer, the TMDs are physically separated, with apparent lipid densities observed in between (Supplementary Fig. 3a). Due to the lower local resolution, we could not unambiguously identify the specific lipids. However, we performed thin-layer chromatography (TLC) to confirm the presence of the bound lipids, which are likely phosphatidylcholine (PC), phosphatidylserine (PS), phosphatidylethanolamine (PE), and cholesterol, without precise identification (Supplementary Fig. 3b). Although these lipids likely contribute to the dimer assembly, the primary mediator of dimerization is the cytosolic region of SLC26A2. The buried interface between the STAS domain in one protomer and the TMD in the other protomer spans ~600 Å², as calculated using ChimeraX[26], and involves strong hydrogen bonds (H-bonds) between specific residues, such as Y247/Q547 and D511/T661, with bond lengths ranging from 2.1 to 3.3 Å (Fig. 1c). In addition, the two STAS domains are attached through hydrophobic interactions, with a buried surface area of ~1100 Å². Specific interactions found include H-bonds between Q657 and T546 and the backbone of R545, H-bond between R545 and D660, as well as the cation-π interaction between F658 and R545 (Fig. 1c). Interestingly, in previous studies of SLC26A5 and SLC26A9, a universal while critical dimerization factor

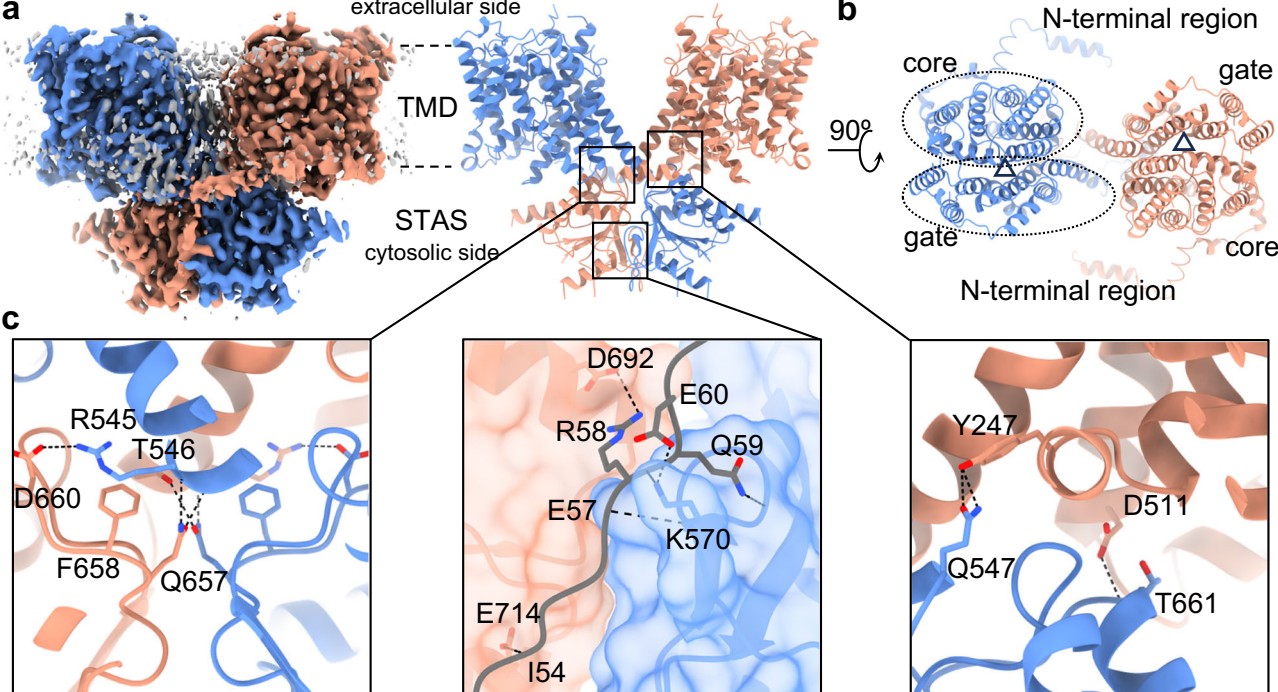

**Fig. 1 | The overall structure of SLC26A2 homodimer. a** The front view of the 3 Å resolution map of SLC26A2-Cl⁻. **b** The final model of SLC26A2 is a domain-swapped homodimer. The two protomers are dark salmon and blue. The lipid/detergent belt is gray. The extracellular side is at the top, and the cytosolic side is at the bottom. The substrate translocation pathways are marked by triangles. **c** The dimerization is mediated by detailed interactions between domains. Specific H-bonds are highlighted. The N-terminal region from the dark salmon protomer is shown in gray for better illustration.

was identified as an anti-parallel β-sheet formed by joining a β-strand from each protomer's N-termini at the bottom of STAS domains[18,20,24]. However, in SLC26A2, such a β-sheet was not observed, suggesting a different dimerization mode within the SLC26 family. In SLC26A2, the N-terminal region plays a vital role in the dimeric assembly through a different mechanism. Specifically, residues 53–61 in the N-terminal loop traverse through the shallow groove formed by the two STAS domains, resulting in a buried interface of ~600 Å². Despite this seemingly shallow interaction, it is remarkably stable, with numerous H-bond interactions involving the residues I54, E57, R58, Q59, and E60 (Fig. 1c). To further demonstrate the significance of the N-terminal loop, we introduced truncation mutations Δ1−45 and Δ1−65. The expression level of Δ1−45 remained at approximately 55% compared to the wild-type, while Δ1−65 was nearly undetectable (Supplementary Fig. 4). This suggests that N-termini-mediated dimerization, especially involving residues 53–61, is critical for the stability of SLC26A2.

## Inward-facing conformation

The substrate transport mechanism of SLC26A2 follows the classic "alternating access" model, where the protein alternates between major conformations, including inward-facing, occluded, and outward-facing. In each conformation, the substrate-binding pocket is solvent accessible to only one side of the membrane bilayer[27]. To determine the specific conformation of SLC26A2, we compared our SLC26A2-Cl⁻ structure with known homologous structures, namely inward-facing SLC26A9[24], occluded SLC26A5[20], and outward-facing SLC4A1[28]. Previous studies have shown that an elevator-like motion between the core and gate domains in TMDs governs the conformational changes in these transporters. This motion is reflected in the vertical position of TM3/TM10 from the core within the lipid bilayer, as it corresponds to the location of the substrate-binding pocket (Supplementary Fig. 2). Here, we aligned the gate domains in all structures and compared their core domains. We found that the vertical position of TM3/TM10 in SLC26A2 closely resembles that of inward-facing SLC26A9 (Fig. 2a & Supplementary Fig. 5a). Using MOLEonline[29], we calculated the pore size in available SLC homologs (Supplementary Fig. 5b). We considered the pore to be open when its radius is larger than the theoretical radius of a Cl⁻ ion without hydration, which is ~1.8 Å. The result shows that our SLC26A2 structure is open to the cytosolic side and closed to the extracellular side. In addition, we performed molecular dynamics simulations using our SLC26A2 models. All simulations, lasting 1 μs, showed that the overall structure remained unchanged, and the substrate-binding pocket between TM3/TM10 became filled with water

from the cytosolic side (Fig. 2b). These results collectively suggest that SLC26A2 adopts an inward-facing conformation.

## Multiple substrate-binding sites

Like other known SLC homologs, the canonical substrate-binding pocket in SLC26A2 is located within the cleft between TM3 and TM10, where the two helices face each other with their N-terminal ends head-to-head (Fig. 2a). Notably, in the structure of SLC26A2-Cl⁻, we identified a distinct density corresponding to the Cl⁻ ion in this cleft (Fig. 3a). Interestingly, this Cl⁻ does not interact strongly with any surrounding residues. Within 5 Å distance from the Cl⁻, there are five hydrophobic residues: Y129 from TM1, V167, G166, F165 from the end of TM3, and L491 from TM12. The primary binding force for Cl⁻ appears to be the weak TM3 and TM10 helical dipoles with positive charges pointing towards the substrate-binding pocket. To further understand the Cl⁻ binding, we performed molecular dynamics simulations, each lasting 1 μs. These simulations revealed that the Cl⁻ ion frequently moves in and out of the substrate-binding pocket, as evidenced by its distance to the end of TM3 (residue G166) (Supplementary Fig. 6). When such distance is between 3.5 and 8 Å, the Cl⁻ ion is within the TM3/TM10 cleft and considered bound. Combining data from three individual runs, we estimated that the substrate-binding pocket is occupied by Cl⁻ ~82 ± 4% of the time.

To understand how SLC26A2 interacts with its primary substrate $SO_4^{2-}$, we determined the structure of SLC26A2-$SO_4^{2-}$. Specifically, in the last purification step, SLC26A2 was eluted in the buffer containing 50 mM $Na_2SO_4$ without NaCl. The final structure of SLC26A2-$SO_4^{2-}$ reached an overall resolution of ~3.5 Å, with a clear and high signal-to-noise ratio density of $SO_4^{2-}$ in the substrate-binding pocket (Fig. 3b & Supplementary Fig. 7). It is noteworthy that, despite the proximity of Cl⁻ to TM3 in SLC26A2-Cl⁻, the $SO_4^{2-}$ in SLC26A2-$SO_4^{2-}$ is instead closer to TM10, approximately 2 ~ 3 Å away from the Cl⁻ binding site. Since $SO_4^{2-}$ is still in the TM3/TM10 cleft, the substrate is stabilized by weak helical dipoles just as Cl⁻. However, this apparent shift in the binding site creates a different local environment surrounding the substrates. Within the 5 Å radius of $SO_4^{2-}$, we found several residues including Y129, Q225 from TM1, F165, I164 from the end of TM3, N490, L491 from TM12, and A439, L440, A441, K442 from TM10. These residues, particularly Y129, K442, Q225, and N490, are likely to interact with $SO_4^{2-}$, providing additional stabilization forces. Indeed, our molecular dynamics simulations on SLC26A2-$SO_4^{2-}$ demonstrated that $SO_4^{2-}$ remains tightly bound to the binding site 100 ± 0% of the time without any significant movements (Supplementary Fig. 6).

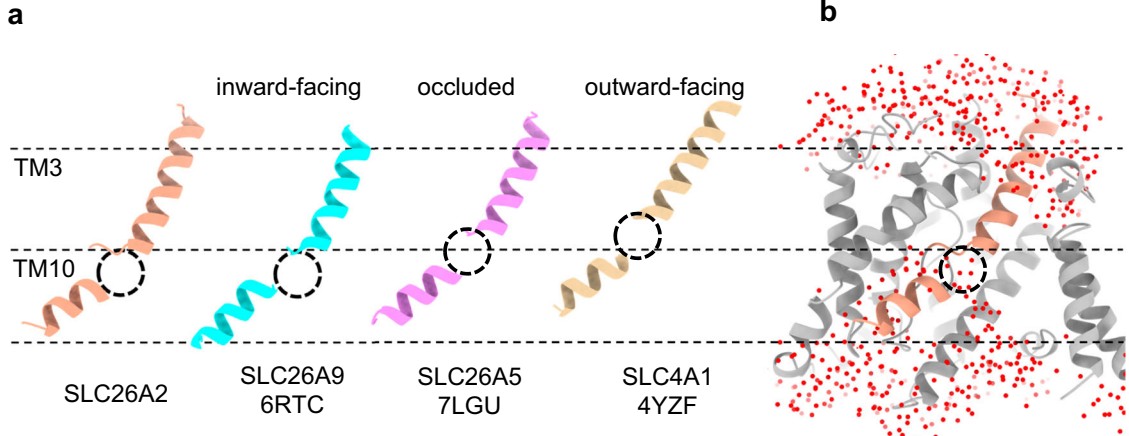

**Fig. 2 | Inward-facing SLC26A2. a** The comparison of vertical positions of TM3/TM10 in the membrane bilayer from representative homologous structures. **b** During simulation, the substrate-binding pocket in SLC26A2 (slabbed front view) is filled with water (red ball) from the cytosolic side. Dashed lines indicate the boundaries and center of the lipid bilayer. Dashed circles mark the substrate-binding pockets. SLC26A2 is in gray with TM3/TM10 in dark salmon.

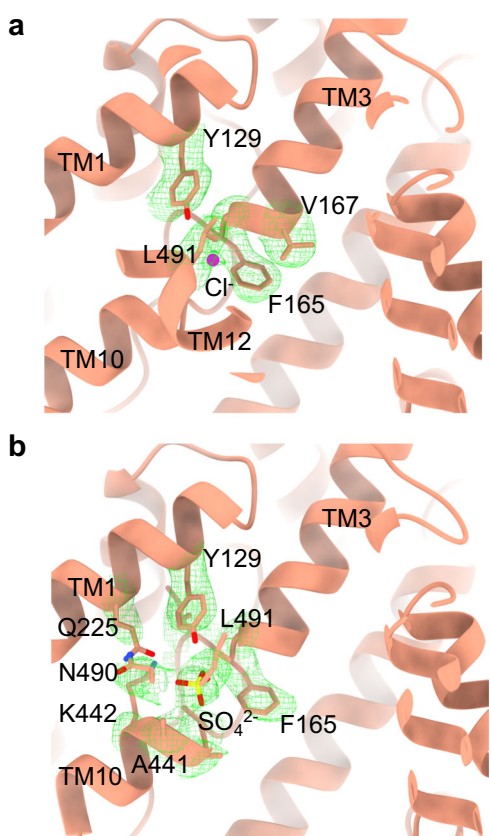

**Fig. 3 | Substrates binding. a** Cl⁻ in the binding pocket between TM3 and TM10, the same view as in Fig. 2. **b** SO₄²⁻ in the binding pocket.

In addition to its known transport of $SO_4^{2-}$ and $Cl^-$, SLC26A2 is also capable of $C_2O_4^{2-}$ transport. We determined the structure of SLC26A2-$C_2O_4^{2-}$ without NaCl to an overall resolution of ~3 Å (Supplementary Fig. 8). Surprisingly, we found no experimental density for $C_2O_4^{2-}$ in the canonical substrate-binding pocket between TM3 and TM10. Instead, we found weak densities outside the pocket, in the core and gate domains interface between TM5 and TM12, which could accommodate a $C_2O_4^{2-}$ molecule (Supplementary Fig. 9). To assess the agreement between the modeled $C_2O_4^{2-}$ and experimental densities, we calculated the Q-scores using ChimeraX, which should be ~0.6 at the reported resolution of 3 - 3.5 Å[30]. The result shows that $C_2O_4^{2-}$ scored 0.77, confirming the model's reliability. Our model shows that the $C_2O_4^{2-}$ substrate is likely stabilized via interactions with the side chains of adjacent Ser residues, as well as backbone carbonyls from L253. We mutated the Ser residues and carried out the cellular transport assay. The result shows a ~40% reduction in transport efficiency, confirming the importance of these Ser residues for $C_2O_4^{2-}$ translocation (Supplementary Fig. 10a). In addition, we performed molecular dynamics simulations to probe the dynamics between SLC26A2 and the $C_2O_4^{2-}$. The result showed that $C_2O_4^{2-}$ remains bound for ~65 ± 4 % of the time. Notably, in all simulations, the $C_2O_4^{2-}$ molecule does not enter the exact binding pocket that accommodates $Cl^-$ and $SO_4^{2-}$ (Supplementary Fig. 6d), implying that the substrates' transport pathways might not be exactly the same.

### Mapping the pathogenic mutations

According to the ClinVar database (accessed August 2023), nearly 200 mutations in the *SLC26A2* gene have been documented with a clinical significance of either pathogenic or likely pathogenic. Among these mutations, 25 are missense mutations capable of producing proteins. We successfully mapped all these missense mutations, involving 21 residues, onto our high-resolution structures (Supplementary Table 2).

How do these mutations cause functional problems for SLC26A2? After careful analysis, we could classify them into several categories based on their potential effects. The first category of mutations most likely affects the overall folding of the transporter, resulting in unstable proteins (Fig. 4a). For example, mutations such as A133V, C311R, A386V, A461V, G484D, and S522F in the TMD replace small hydrophobic side chains with larger and even hydrophilic ones, leading to potential disruption in helical packing. For instance, A386V expression in *Xenopus* oocytes was reported to be significantly reduced compared to the wild-type[14]. Moreover, residues like Gly and Pro, found in A386G (in TM8) and L483P (in TM12) mutants, have been known to disrupt α-helices, contributing to their functional problems[31]. The substitution of N425 with negatively charged Asp, found in the N425D mutant, leads to H-bond loss with T435, T436, and Q233, destabilizing the loop containing T435 and T436 (Supplementary Fig. 11a). All these changes are energetically unfavorable as the whole region is within the membrane bilayer. In the STAS domains (Fig. 4b), mutations like G678V, A715T, C653S, C653Y, and C653G may impact protein functionality. G678 sits at the very short turn between an α-helix and β-strand, which is apparently required for such an arrangement of secondary structures and should favor Gly over G678V mutation. The side chain of A715 disallows hydrophilic mutations like A715T as it points towards a highly hydrophobic environment formed by residues from the β-sheet and adjacent loop (V650, L682, V712, and F709) (Supplementary Fig. 11b). C653 is at the end of a β-strand, closely surrounded by hydrophobic residues I651, I656, L683, and C686. The side chain of C653 forms a weak H-bond with the backbone of I651 (Supplementary Fig. 11c). C653S is pathogenic because Ser is inherently more hydrophilic than Cys and is harder to tolerate the hydrophobic environment surrounding C653. C653Y is problematic because of the large size of the side chain, while C653G is unfavorable because of the intrinsic destabilizing nature of Gly in the β-sheets[32]. Although Gly destabilization can be 'rescued' by specific cross-strand pairing with aromatic residues, no aromatic residues are present around this region.

The second category of mutations in the *SLC26A2* gene most likely impacts protein-lipid interactions (Fig. 4c). For instance, I426 is located at the midpoint of the lipid bilayer, faces toward the surrounding environment, and thus hydrophobically interacts with lipid tails. Mutations like I426T and I426N would be unfavorable because the side chains are much more hydrophilic. Mutations like D111Y, S157P, R279W, Q454P, W505R, and T512K have reversed charge properties compared to the wild-type, which could affect their interactions with the lipid headgroups because these residues are all predicted to be close to the lipid boundaries by the PPM3 server based on our structures[33]. Among these mutations, R279W is the most common pathogenic variant found outside of Finland[34]. In *Xenopus* oocytes, R279W shows a similar surface abundance as the wild-type but exhibits a reduced substrate transport rate, suggesting it is a partially dysfunctional mutant[14]. In contrast, another study using HEK293 cells demonstrated that R279W is expressed ~50% less on the cell surface, and these cells have ~50% reduced rates of sulfate transport compared to cells expressing the wild-type, suggesting that the transport efficiency of R279W mutant after normalization is roughly at the same level of wild-type SLC26A2[8]. In our structural model, R279 is located on an extracellular loop away from the potential substrate translocation path, making it difficult to impact the substrate translocation directly. Furthermore, in our cellular transport assay, we observed a ~50% reduced transport and ~55% reduced expression level of the R279W mutant compared to the wild-type, which translates to indistinguishable transport efficiency after normalization for the mutant protein compared to the wild-type protein (Supplementary Fig. 10b). This finding aligns with the latter study, suggesting that the R279W mutant may retain its normal substrate transport function. Instead, the mutant may lose its natural lipid anchoring ability, making it harder to insert

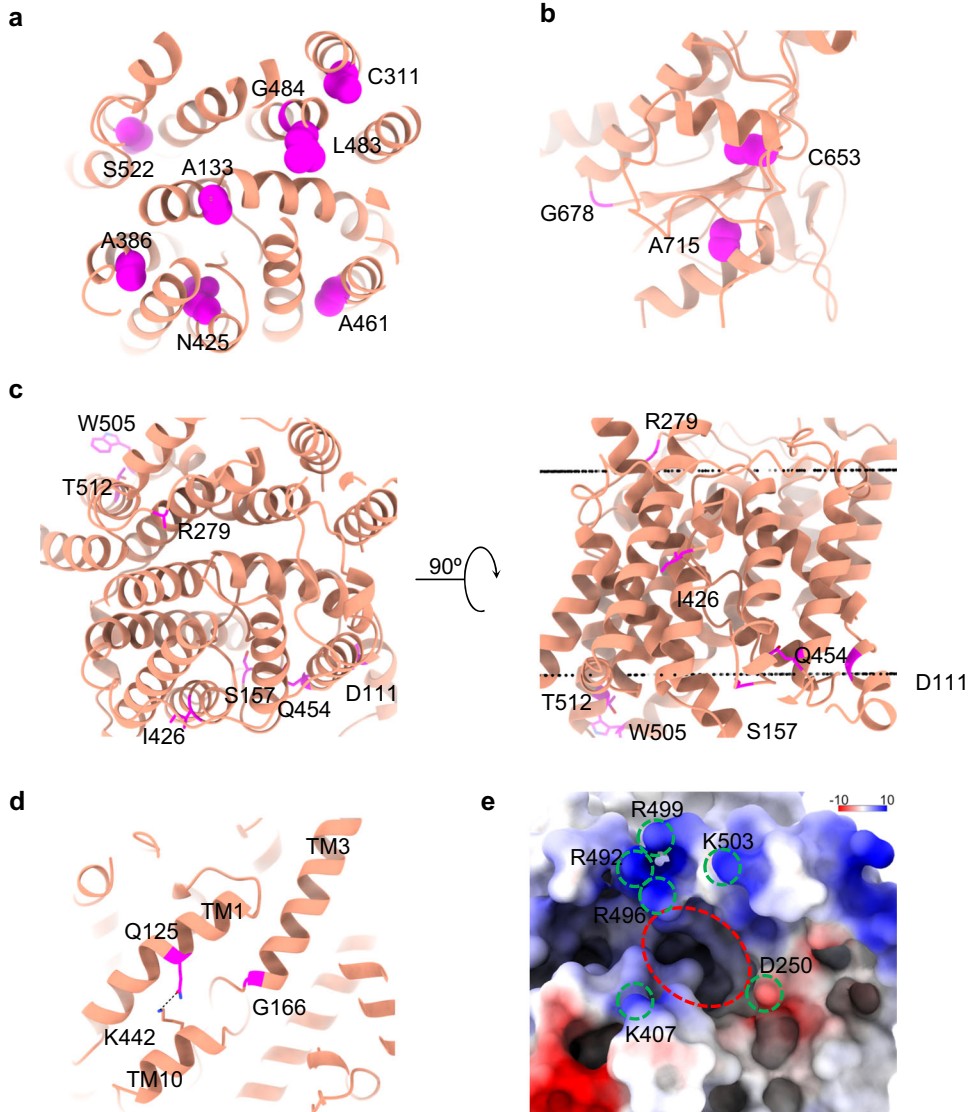

**Fig. 4 | Pathogenic mutations mapped on the SLC26A2 model. a** Residues in TMDs that affect the protein stability. **b** Residues in soluble STAS domains that affect the protein stability. **c** Residues that may be critical for lipid interactions. The dashed black line represents the predicted boundary of the lipid bilayer. **d** G166 and Q125 around the TM3/TM10 cleft. **e** Electrostatic surface representation of the TMD domain viewing from the cytosolic side. The dashed red circle represents the putative transport pathway, while the green ones mark charged residues.

into the membrane correctly. This results in a reduced expression level and, thus, impaired substrate translocation.

The third type of mutation directly affects the interactions between substrates and the transporter, as they are around the substrate-binding pocket. For instance, Q125 is located in TM1, seemingly away from the TM3/TM10 cleft (Fig. 4d). However, it forms an essential H-bond with K442 in TM10, the only strong interaction stabilizing the K442 side chain. We hypothesized that the Q125L mutation would release the K442 side chain, allowing it to insert into the TM3/TM10 cleft easily. Similarly, G166 defines the edge of the substrate-binding pocket at the N-terminal end of TM3, and the G166R mutation would place its side chain into the substrate-binding cleft. Due to the positive charge of K442 and G166R, these residues may interact strongly with negatively charged substrates. Such strong interactions could impede substrate release and translocation, creating unfavorable conditions for the transport process. Such a hypothesis was tested by cellular transport assay using the following single mutations: Q125L, G166A, and G166R (Supplementary Fig. 6). The result shows that Q125L and G166R retain approximately

40 - 70% of sulfate translocation, while G166A has the same transport efficiency as the wild-type.

Notably, D250 is situated away from any lipids and is located at the end of TM5, near the cytosolic entry of the substrate translocation pathway (Fig. 4e). We analyzed the electrostatic distribution of the surface around the area using ChimeraX. As expected, the cytosolic entry is surrounded by positively charged residues, including K407, R492, R496, R499, and K503. This positive charge distribution seems essential to accommodate negatively charged substrates under neutral pH conditions. Interestingly, D250 is the only negatively charged residue in this area and the only one associated with pathogenicity identified so far (D250V). Our cellular transport assay shows that D250V has almost the same expression level and translocation ability as wild-type (Supplementary Fig. 10b). Thus, why D250V is pathogenic could not be explained in this study and needs further investigation.

## Discussion

In this study, we have determined three distinct structures of SLC26A2 transporter in complex with substrates $SO_4^{2-}$, $Cl^-$, and $C_2O_4^{2-}$. Despite

adopting a similar inward-facing conformation, these structures showcase unique binding modes for different molecules. Prior studies have already established that the primary role of SLC26A2 is to transport $SO_4^{2-}$, even though it can effectively transport a variety of anions[35,36]. Our molecular dynamic simulations further underscore this idea. Among the small molecules studied, $SO_4^{2-}$ is the only one consistently staying in its original binding site, indicating a relatively more robust substrate-transporter interaction (Supplementary Fig. 6). By analyzing the total time substrates spend in their binding site during simulations, we notice an order of binding affinities from high to low: $SO_4^{2-} > Cl^- > C_2O_4^{2-}$. This order aligns with the documented efflux rate: $SO_4^{2-} \sim Cl^- > C_2O_4^{2-}$, measured under consistent extracellular conditions in oocytes[14].

It is well established that the anion exchange mediated by SLC26A2 is electroneutral[14,15]. However, the ongoing debate centers on whether the exchange mechanism operates as $SO_4^{2-}/2Cl^-$ or $SO_4^{2-}/OH^-/Cl^-$. Do our structural studies shed light on this topic? First, we do not consider the exchange mode involving proton, such as $SO_4^{2-}/H^+/Cl^-$, because our structures show no protonatable residues (Glu and Asp) near the TM3/TM10 cleft region nor the potential translocation pathway. Second, in the context of extracellular $SO_4^{2-}$ exchange with intracellular $Cl^-$, the stoichiometry of one $SO_4^{2-}$ exchanging with two $Cl^-$ would require simultaneous binding of two $Cl^-$ ions in the TM3/TM10 cleft. This scenario appears highly unlikely for the following reasons: a) only a single $Cl^-$ density was observed in the SLC26A2-$Cl^-$ structure; b) in the SLC26A2-$Cl^-$ simulations performed in the presence of 100 mM KCl, we observed only 0 or 1 ions binding in the cleft on the timescale of the trajectories. Thus, to remain electroneutral, the most plausible scenario should involve co-transporting one $OH^-$ and one $Cl^-$ for exchanging a single $SO_4^{2-}$. Given the resolution of our structures, it is challenging to discern specific densities for individual $OH^-$. However, hydrophilic residues in the binding pocket, such as Y129, K442, Q225, and N490, could likely play a role in $OH^-$ translocation.

Upon comparing all three structures, we noticed that the size of the substrate-binding pocket within the TM3/TM10 cleft slightly changes in response to different substrates. To quantify that, we measured the distance between TM3 and TM10 using Cα positions of three pairs of residues: F165/K442, G166/A441, and V167/L440 (Supplementary Fig. 12a). The analysis reveals that the cleft size is at its smallest when bound to $SO_4^{2-}$, intermediate when bound to $Cl^-$, and largest when $C_2O_4^{2-}$. However, the variation results in only a maximum change of approximately 1-2 Å in diameter. Recent studies on SLC26A5 have reported similar size changes within the TM3/TM10 region in response to the binding of small molecules like $SO_4^{2-}$, $Cl^-$, or the inhibitor salicylate. The authors of those studies propose that these changes correlate with a larger-scale global conformational shift between expanded and contracted states[19,20]. Interestingly, the cleft change observed in SLC26A5 is at the same scale as observed in our SLC26A2 structures. It is also important to note that the distance measurements at the reported resolution (3-3.5 Å) are typically associated with a margin of error, which could be as large as 0.5 Å[37]. Furthermore, all our SLC26A2 structures align remarkably well with each other with an RMSD of ~0.8 Å (Supplementary Fig. 12b). Taken together; we believe such minute changes in the pocket size likely have no significant impact on SLC26A2. Such a disparity may arise from the dual nature of SLC26A5 as both a voltage-sensing motor and a transporter, in contrast to SLC26A2's sole transporter function[38]. After all, the precise mechanism governing the transition from the inward-facing to the occluded and outward-facing conformation of SLC26 members remains a topic for future research.

Here, with the high-resolution structures and related MD simulations, we could shed some light on the $SO_4^{2-}/Cl^-$ exchange mechanism of SLC26A2. First, $SO_4^{2-}$ and $Cl^-$ ions bind to the same substrate-binding pocket within the TM3/TM10 cleft, though their interactions with surrounding residues are not the same. This phenomenon is not unprecedented. For example, in the nitrate/nitrite exchanger NarK, both substrates occupy essentially the same pocket formed in between two Arg residues, R305 and R89, with R89 playing a more pivotal role in the binding of nitrite over nitrate[39,40]. Similarly, in the electroneutral $Na^+/H^+$ antiporter NhaP, both $Na^+$ and $H^+$ bind to the same pocket around D159 but with distinct interactions with surrounding residues[41]. Second, the SLC26A2-$SO_4^{2-}$ structure in the inward-facing conformation most likely represents the state post-import of extracellular $SO_4^{2-}$. Thus, the stabilization of $SO_4^{2-}$ in the binding pocket in all independent simulations suggests a mechanism that prevents premature leakage. The question then arises as to how the imported $SO_4^{2-}$ is released from the substrate-binding pocket. Comparing the simulations of SLC26A2-$Cl^-$ and SLC26A2-$SO_4^{2-}$, we found that $Cl^-$ ion can intersect the $SO_4^{2-}$ binding site when within 6-8 Å of G166, implying a possible competition between the two substrates. We previously noted that $SO_4^{2-}$ appears to have a stronger binding affinity, given its longer residency within the binding pocket. Nevertheless, considering that transporter turnover rates are typically several to several hundred per second[42,43], the 1 µs duration of our simulations is likely too short and thus biased. We hypothesize that, over extended timeframes, intracellular $Cl^-$ may competitively bind to the TM3/TM10 cleft, facilitating the release of the imported $SO_4^{2-}$. Third, our structures revealed that while $SO_4^{2-}$ and $Cl^-$ share a binding pocket, the $C_2O_4^{2-}$ ion binds to a distinct site and does not interact with the $SO_4^{2-}/Cl^-$ pocket during all simulations, implying alternative translocation pathways for different substrates. Such a phenomenon has been documented before. For example, in *Vibrio cholerae*, the multidrug and toxic compound extrusion transporter NorM uses two distinct ion-translocation pathways for Na+ and H+ transport[44]. Our finding suggests that the molecular mechanism governing $SO_4^{2-}/Cl^-$ exchange may differ from that of the $C_2O_4^{2-}/Cl^-$ exchange. To delineate the mechanistic details, future work will require additional structural determination of SLC26A2 in alternative conformations alongside more sophisticated MD simulations.

## Methods

### Protein expression and purification

The full-length human SLC26A2 was cloned into a pEZT-BM (Addgene) vector with an N-terminal His-tag and a thrombin digestion site. The resulting construct was overexpressed in Expi293F cells (Thermo-Fisher Scientific) using the BacMam Expression System. After 72 h of virus infection, the cells were harvested through centrifugation and lysed by being passed three times through a microfluidizer M110P (Microfluidics Corporation). The membrane fraction was collected by ultracentrifugation at 150,000 g for 1 h and resuspended in buffer A (20 mM HEPES pH 7.5 and 150 mM NaCl). To solubilize SLC26A2, the membrane was incubated with 1% lauryl maltose neopentyl glycol (LMNG) (Anatrace) at 4 °C for 2 hrs. The supernatant containing the target protein was isolated by ultracentrifugation at 1,50,000 g for 1 hr and then incubated with TALON IMAC resin (Clontech) in buffer A with 5 mM imidazole. After washing the resin with buffer B (20 mM HEPES pH 7.5, 150 mM NaCl, 0.003% LMNG) containing 10 mM imidazole, SLC26A2 was eluted in buffer B with 200 mM imidazole. The eluted protein was further digested with thrombin (Enzyme Research Laboratories) at a molar ratio 1:50 overnight at 4 °C. Subsequently, SLC26A2-$Cl^-$ was purified by gel filtration chromatography with a Superose 6 column (Sigma-Aldrich) in buffer B. To prepare SLC26A2-$C_2O_4^{2-}$, NaCl in SLC26A2-$Cl^-$ was exchanged with 50 mM $K_2C_2O_4$. To prepare SLC26A2-$SO_4^{2-}$, NaCl in SLC26A2-$Cl^-$ was exchanged with 50 mM $Na_2SO_4$.

### Cryo-EM data collection

The freshly purified SLC26A2 samples were concentrated to ~3.5 mg/ml. 3 µl of each sample was applied to a plasma-cleaned UltrAuFoil gold grid (R 1.2/1.3, 300 mesh, Electron Microscopy Sciences) and prepared using a Vitrobot Mark IV (Thermo Fisher Scientific) with the

environmental chamber set at 100% humidity and 4 °C. The grid was blotted for ~1.5 s and then flash-frozen in liquid ethane. The data was collected on a Titan Krios (Thermo Fisher Scientific) operated at 300 keV and equipped with a K3 direct detector (Gatan) in the Pacific Northwest Center for Cryo-EM (PNCC). The detailed parameters for data collection are shown in Supplementary Table 1.

### Image processing, model building, and model refinement
All cryo-EM data were processed using cryoSPARC v4[45]. Specifically, Patch Motion Correction was used to correct the beam-induced movement, and Patch CTF was used to estimate contrast transfer function parameters for each movie. Particles were picked with the help of Topaz and extracted with a box size of 320 × 320 pixels[46]. After several rounds of reference-free 2D classification, ab initio reconstruction, and heterogeneous refinement, the best particles were selected for further processing. The final reconstructions were obtained with 100k~150k particles at the resolution of 3.2~3.6 Å. Model building of SLC26A2 was performed in Coot[47]. AlphaFold predicted model of SLC26A2, together with several homologous structures available in the Protein Data Bank (PDB), were used as a guide[48]. The final models were refined in PHENIX[49]. The quality of the models was assessed using MolProbity[50]. Statistical details are provided in Supplementary Table 1. All figures, movies, and charts were prepared using UCSF ChimeraX[26] and Microsoft Excel.

### Molecular dynamics simulations
The missing loops and side chains in all SLC26A2 models (residues 52–724) were built using the Molefacture plugin in VMD[51]. The protonation states of titratable residues were assumed to be those at pH7 by PROPKA[52]. The coordinates of the bound substrates, including $Cl^-$, $SO_4^{2-}$, and $C_2O_4^{2-}$, were based on the modeled ions in our cryo-EM structures. Molecular models for all MD simulations were further prepared using the online tool CHARMM-GUI[53]. Specifically, the proteins with substrates were embedded in a lipid bilayer made of POPC, whose orientation and position were calculated by the Orientations of Proteins in Membranes (OPM) server[54]. Parameters with all substrates were generated by the CHARMM general force field (CGenFF)[55,56]. The systems were solvated with TIP3P water, including 100 mM KCl, resulting in a periodic box dimension of ~140 × 140 x 160 Å³, containing ~250,000 atoms.

All simulations were performed using GROMACS[57]. The force field parameters of the protein and lipids were CHARMM36m and CHARMM36, respectively[58,59]. The grid information for PME (Particle-Mesh Ewald) electrostatics was generated automatically. A constant number of particles (N), pressure (P) of 1 bar, and temperature of 310 K were used for the NPT ensemble. A 2 fs timestep was used. All simulations were carried out with the following steps: 1) 5000 steps of energy-minimization; 2) 20 ns of equilibration with gradually decreasing positional restraints; 3) 1 μs of production MD simulation with three repetitions starting from the same equilibrated systems after the Step 2.

### Thin-layer chromatography
SLC26A2 samples were subjected to two rounds of purification using the Superose 6 column to remove non-specifically associated lipids. The TLC experiment was performed following a previously described method[60]. Specifically, the lipids tightly associated with SLC26A2 were extracted using a mixed solution of chloroform and methanol (2:1). A TLC plate (Millipore Sigma) was washed with ethanol, air dried, and then activated at 100 °C for 30 min. SLC26A2 samples and lipid standards were dotted on the TLC plate. The experiment was performed with a first mobile phase of chloroform-methanol-water (65:25:4, v/v/v) till the solvent front was halfway through the plate, followed by a second mobile phase of hexane-acetone (100:1, v/v) till the solvent front reached the top. After the run, the TLC plate was air dried,

stained with 0.03% (w/v) G-250 in 20% methanol for 15 min, destained in 20% methanol for 10 min, and then air dried again.

### Cellular transport assay
Expi293F cells expressing wild-type and various mutations of SLC26A2 were used for the sulfate uptake experiments. Specifically, Expi293F cells were infected with corresponding viruses for 24 h to allow protein expression. $0.2 × 10^6$ cells were washed twice with Wash Buffer (240 mM mannitol, 2.5 mM potassium sulfate, 2.8 mM calcium gluconate, 1.2 mM magnesium sulfate, and 10 mM HEPES pH 7.5) and then incubated with 400 μl of transport buffer (Wash Buffer plus 3 μCi $^{35}$S-sulfate) for 10 min at room temperature. The reaction was stopped by adding 400 μl of ice-cold wash buffer, quickly followed by filtration using pre-wetted cellulose nitrate filters. The filters were washed with 5 ml of wash buffer, dried for 1 h, and dissolved in 4 ml of Filter Count scintillation liquid (Perkin Elmer). The $^{35}$S radioactivity within the cells was quantified by using a Tri-carb 2910 TR Scintillation counter (Perkin Elmer).

### Reporting summary
Further information on research design is available in the Nature Portfolio Reporting Summary linked to this article.

## Data availability
Pathogenic mutations in the *SLC26A2* gene are accessible from the ClinVar database. The cryo-EM maps of SLC26A2 were deposited in the Electron Microscopy Data Bank under the accession codes EMD-41427, EMD-41428, and EMD-41429. Their corresponding coordinates of the atomic model were deposited in the Protein Data Bank under the accession codes 8TNW, 8TNX, and 8TNY, respectively. All other data are available from the corresponding author upon request. Source data are provided in this paper.

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

## Acknowledgements

A portion of this research was supported by NIH grant U24GM129547 and performed at the PNCC at OHSU and accessed through EMSL (grid.436923.9), a DOE Office of Science User Facility sponsored by the Office of Biological and Environmental Research. This work is partially supported by NIH (R01 GM126626, R01 HL168686, and R21 AI175646).

## Author contributions

W.H., A.S., and H.Z. designed the experiments, collected and analyzed the data, and wrote the manuscript.

## Competing interests

The authors declare no competing interests.
