## [Peer Review File · Nature Communications]

Substrate Binding Plasticity Revealed by Cryo-EM Structures of SLC26A2Reviewers' comments:

Reviewer #1 (Remarks to the Author):

In this paper, the author solved three structures of SLC26A2 in complex with Cl⁻, SO₄²⁻ and C₂O₄²⁻ by cryo-EM. By analyzing the structures, the authors determined a unique dimer interface that is different from other SLC26 family members. By comparison with related transporters, the authors observe that their SLC26A2-Cl⁻ complex forms an inward-facing conformation. Combined with MD simulation, the author illustrated the plasticity of SLC26A2's ligand binding pocket and the molecular details of its binding site bound to different substrates. Finally, the author mapped pathogenic mutations and try to explain the mechanisms underlying their pathogenicity. This paper provides some interesting insight, particularly into the oxalate binding site, however, a lot of the conclusions are not fully backed up by the data, and the section on insights into the mechanisms of pathogenic mutations is very confusing and lacks data.

1. The supplied structural coordinates curiously only contain a single protomer of the SLC26A2 dimer, so no claims regarding the dimer interface (Fig 1) could be evaluated.

2. The binding pose(s) of oxalate are very intriguing, and perhaps the most interesting finding reported in the manuscript. However, both oxalate molecules are built into somewhat ambiguous and weaker density, and Figure 3C indicates as much. Although the authors perform MD simulations to attempt to validate the binding locations, these are based on the built SLC26A2-oxalate complexes and thus only indirectly validate the binding poses. It can therefore not be completely ruled out that these densities correspond to e.g. other buffer components, though I am inclined to believe the authors. Nonetheless, these unusual but very intriguing binding modes require additional validation, and I am suggesting that the authors perform oxalate transport experiments. Testing different substitutions for S245, S399, or L253 in the vicinity of oxalate should affect oxalate transport or dissociation of transporter-substrate complexes if the proposed binding sites are correct.

3. Chloride, on the other hand, does not fit at all into the provided density and Figure 3A shows as much. However, there is clear density for chloride in the vicinity of the current position, and this could be remedied by simply moving the ion. The pocket for this ion is generally not very well built, and key residues such as Q125 need to be better fit into the density. Overall, these issues should be comparatively easy to fix and would much improve the quality of the model. That being said, that does not solve the issue of how chloride is actually coordinated as the authors already state that they do not observe any strong interactions with surrounding residues. Are there potential OH ions involved? Have the authors tested that in MD simulations? How does this compare to Chloride or anion binding sites in other SLC transporters? The chloride must be coordinated somehow as it is currently just "floating in space".

4. The authors functionally characterize pathogenic mutations in “Mapping the pathogenic mutations” section. This section is very confusing. For instance, only selected mutations have been characterized. What is the justification for only testing the mutations shown in Figure S5? The authors also mention a list of mutations in Line 215/216 that they claim likely affect transporter folding. Shouldn't this be tested by ELISA or Western blotting? The authors then say that a R279W mutant has previously been shown to exhibit 50% reduced surface expression and a 50% reduced transport rate, but then state that this shows that the mutant 279W does not dramatically affect transport function (Line 250/251). I would consider a 50% reduction in function and trafficking dramatic for sure. But then it gets even more confusing. The authors next test this mutant themselves and state that they also observed “a ~50% reduced transport and ~55% reduced expression level of the R279W mutant compared to the wild-type (Figure S5)” (Line 253/254). First, Figure S5 doesn't even show any expression level data, and 55% is an oddly precise number for any expression level determination. Second, the R279W mutant doesn't even show the claimed 50% reduction in transport, as the transport efficiency seems comparable to wt according to Figure S5. Also Figure S6 is mentioned in the text before Figure S5.

5. How do the authors arrive at the claim that the “expression level of $\Delta 1-45$ remained at approximately 50% compared to the wild-type”? Figure S3 shows a pretty rough looking western blot/gel (not clear since not described), and it is unknown how the 50% was calculated. Maybe ELISA would be better to more accurately quantify surface expression levels. Also, there seem to be large aggregation spots on the blot/gel. Is it possible that the expression levels of wt and $\Delta 1-45$ are similar, but the n-terminally truncated construct aggregates during extraction or exposure to detergent in the gel running buffer?

6. The authors analyze cleft sizes in Figure S7 and line 312 forward. Some of these distances such as G166/A441 are less than 0.4 Å. Given the resolution of >3 Å for the structures, there is considerable error associated with atom positions in the models (usually around 0.3-0.5 Å for a 3 Å structure). The authors need to acknowledge that their calculated differences in binding cleft size have an error associated that, in this case, substantially affects the outcome.

7. The title definitely needs improvement and be focused on a big finding(s) rather a generic description

8. In line 114, the authors state: “The two sub-domains undergo an elevator-like rigid-body movement”, I'd like to see this “movement” in figures.

9. In line 118, the authors state: “Due to the lower local resolution, we could not unambiguously identify the specific lipids”. I don't think the discussion of lipids that are not unambiguously identified adds much to the paper. If the authors choose to include it, they should also show densities in a supplementary

figure and discuss the positions and findings from the TLC studies in light of observations made for other SLC26 or related transporters for which lipids have been identified.

10. Figure 2 needs to be strengthened. Comparison of the vertical position of TM3/TM10 in SLC26A2/SLC26A9/SLC26A5/SLC4A1 needs to be expanded to show full protomers. It would also be good to see the differences among the channel pores across the 4 transporters. In addition, the authors need to compare the entrances at both the extracellular and intracellular sides among these 4 transporters to showcase differences and similarities, and better highlight an inward facing state.

11. Figure S1 only shows the processing workflow for one structure. Workflows for all three structures are needed.

12. In Figure S7, please change the colors of the structures to make them easier to distinguish.

13. Line 317: "Recent studies on SLC25A5 have reported..." Do the authors mean SLC26A5?

Reviewer #2 (Remarks to the Author):

Sulfate is an essential nutrient for many organisms, serving as a crucial component in various biological processes, including the synthesis of amino acids, coenzymes, and other biomolecules. Sulfate transporters play a vital role in maintaining the cellular supply of sulfate, ensuring the normal functioning of these processes. The manuscript "Cryo-EM Structures of SLC26A2 Binding with Substrates" by Hu et al. describes cryo-EM structures of the human sulfate transporter in inward-facing conformational states, bound to substrates.

The authors present a valuable research study, however, there is a major issue with this manuscript; the structural analysis of the solved structure should be more careful to avoid any mislead results. Also, the reviewer recommends the authors to make a structural comparison of the substrate with the previous studies in SLC26A.

The reviewer also would like to point out some general issues that need to be considered before resubmission:

Major Points:

The SO4 is of good quality. However, the position of Cl⁻ should be adjusted since it does not align correctly with the density. The presence of C2O4²⁻ is likely the result of noise and should be examined carefully. As you can observe, the position of C2O4²⁻ is not structurally reasonable (no positive charged residues), and there is also similar noise density in the structure of SLC26A2-SO4²⁻.

p165-170: The description for Cl⁻ should be updated after the model adjustment. Since the density is not of high quality, the author should consider using "potentially corresponds to Cl⁻".

p170-174: The MD simulation should be rerun based on the updated model of SLC26A2-Cl⁻.

p185-189: The stabilization of SO4²⁻ is achieved through the interaction with the positively charged end of the transmembrane (TM) dipoles of both TM3 and TM10. The recognition of SO4²⁻ by this anion-dipole interaction should be described in the manuscript.

p195-205: The densities likely result from noise, and the MD simulation does not support the assignment. If the author cannot provide additional evidence, it is advisable to remove the structure from the main manuscript or relocate it to the supplementary material.

Additionally, the discussion should be updated accordingly.

Minor Points:

1) Figure S5: Transport efficiency of SLC26A2 single mutants. No control was provided.

2) I recommend using common names rather than systematic names for SLC26A2 and other SLC proteins. For instance, you can refer to SLC26A2 as "Sulfate transporter" to make the text more reader-friendly and accessible.

3) For better visualization by readers, the deposited structure should be in dimer format. In cryo-EM, complexes with symmetry are typically built with the entire complex.

4) Fig 1A should depict the membrane, extracellular side, and cytosol side.

5) Fig1C (p125-130) :

Fig1C left panel:

The cation- π interaction between R545 and F658 appears to contribute significantly beyond what is shown here. Additionally, the charge-charge interaction between R545 and D660 suggests the need to adjust the direction of the side chain of D660.

Fig1C middle panel:

The figure should include the representation of the main chain for hydrogen-bond; I54 does not form any hydrogen bonds with other residues, although one is depicted here. The side chain of Q is not centered within the density.

Fig1C Right panel:

6) The orientation of the side chain of S538 should be adjusted in certain structures where it cannot form a hydrogen bond.

7) Fig2: The position of the reference domain should be included to illustrate the relative motion of TM3 and TM10.

8) AF models were used in the model building, indicating that it was not a de-novo model building process (p102)

9) "Specifically, residues 53-61 in the N-terminal loop traverse through the shallow groove formed by the two STAS domains, resulting in a buried interface of $\sim 600 \text{ \AA}^2$." This should refer to the calculation software.

Reviewer #3 (Remarks to the Author):

In their paper Hu et al present the first resolved structures of the anionic exchanger SLC26A2 in complex with sulfate, chloride or oxalate. The structures reveal a unique mode of homodimer formation relative to other members of the family with known structures, and complementary simulations show differences in the apparent affinities of binding of the three substrates. Based on the new structural information the authors map out known pathogenic mutations in the SLC26A2 gene and discuss some possible mechanisms for their effect on transport activity.

The paper is easy to read and is well organized. However, many details related to the computational analysis are missing and many of the discussed differences between substrate binding modes are below

the resolution of the structures calling into question the significance of the results and their interpretation. These and some other minor comments are outlined below.

- There are three types of simulations performed in this study, corresponding to the three resolved structures of SLC26A2 in complex with chloride, sulfate and oxalate ions. It is clear that the simulations were run for 1 microsecond each with the Charmm36 force field, and that three replicas (independent runs) were performed, however beyond that there is very little information given. Critical evaluation of the results requires both: (1) more technical details, as well as more robust descriptions of (2) the construction of the systems and (3) their analysis.

(1) what were the simulation parameters used, including temperature; was any additional parameterization of the substrates performed as was done for e.g. sulfate in ref 20; how were the flexible loops missing from the protein structure (Line 104) modeled?

(2) were the substrates initially placed according to the cryo-EM structures, what salt concentration/ions were used, and were individual runs started from the same starting configuration or were they constructed independently from each other? Showing representative snapshots of the simulation trajectories from the beginning and end can help visualize how the protein is sitting in the membrane relative to the lipids, as well as the relative sizes of the membrane patch and water layer (the provided dimensions of the system don't mean much since the size of the protein structure is not indicated).

(3) how similar/different were results from the three replica simulations? For example, what are the errors on the percent time the substrates stayed in their respective binding sites from the independent runs? Was water observed to enter the substrate binding site in all simulations (Line 159)?

- The second category of pathogenic mutations discussed in the text involves residues that "most likely impact protein-lipid interactions" (Lines 238-239). The whole discussion is centered around speculations based on the experimental structure (or what the PPM3 software predicts) but it is surprising that there is no connection to the simulations which provide atomistic resolution into those interactions. Are the known mutations in residues that are in direct contact with lipids in the simulations? How do these contacts evolve during the trajectories, are they dynamic? Showing where the residues are located in a simulation snapshot would be helpful. Can the effect of one or more of these mutations be further confirmed with additional simulations?

- Along the lines of the specific protein-lipid interactions, do the simulations provide any insights into the ability of lipids to stabilize the interactions between the substrate and the protein? How far are lipid tails/headgroups from any of the resolved binding sites? On Line 302 there is a reference to a "potential

translocation pathway". Can this pathway be mapped onto the structure in relation to the surrounding lipids?

- The resolution of the structures is ~ 3.5 Å which is above the reported differences between both the locations of the bound chloride vs sulfate ($\sim 2-3$ Å), and the plasticity of the binding site as measured by changes in pairwise distances in Fig. S7 ($\sim 1-2$ Å). How significant are these differences then? There is no discussion about the uncertainty in these comparisons given the structural resolution.

- Related to the above, the analysis of residues only within 5 Å of the bound substrate in the resolved structures seems very limited. If the chloride ion for example, was sitting closer to where sulfate binds, that would bring it in contact with some positive residues. What are the residue contacts that the substrates make in the simulations? Analyzing their dynamics would be more informative than looking only at the static structures. Similarly, what are the respective distances from Fig S7 and their uncertainties from the simulation trajectories? Is this plasticity of the binding site observed computationally as well?

- The statement on Line 291 that in simulations, sulfate is the only one consistently binding within the TM3/TM10 binding site, implies that the ion was initially in solution and then bound to the binding site which is not what the data in Fig S4 shows. It looks like the ion was initially placed in the binding pocket (as in the structure), and it never came out of it during the 1 μ s simulation. This is different from letting the ion explore different binding sites. The text needs to be edited to remove any ambiguity.

- Similarly, the statement on Lines 303-304 is wrong since no simulations were performed with multiple anions to show that only one binds to the TM3/TM10 cleft.

- Given that SLC26A2 is an exchanger (antiporter), what transport mechanism from known models is consistent with sulfate and chloride binding to the same site in the inward-facing conformation? Is their potentially different affinity for the binding site sufficient to make one come off while the other one binds, or is it possible that there is cooperative behavior as indicated by e.g. the different binding site for oxalate? It is mentioned that different substrates may have different mechanisms of transport but are there such examples in the literature? Discussing these questions would help reconcile the experimentally observed binding modes in the context of the protein function.

- As stated in the abstract, the molecular mechanism of substrate translocation by SLC26A2 has remained unknown because of the lack of structural information. In at least two places in the text however, there are statements implying that the mechanism is known and involves an elevator-like rigid-body movement (Lines 114-116 and Line 145). These statements are contradictory and need to be modified/clarified accordingly.

- In Fig. 2 it is hard to see TM3 and TM10 in the simulation snapshot. It would help if the helices are colored with two distinct colors, and the water molecules are shown transparent to more clearly show the binding pocket.

- All pots on Fig. S4 need axis labels.

We would like to express our sincere gratitude to the reviewers and the editorial team for their time and valuable feedback on our manuscript. We appreciate the insightful comments and suggestions, which have significantly improved the quality of our work. Below, we provide a point-to-point clarification and address each reviewer's comments.

Reviewer #1's comments:

1. The supplied structural coordinates curiously only contain a single protomer of the SLC26A2 dimer, so no claims regarding the dimer interface (Fig 1) could be evaluated.

Response: *The PDB files have been revised to include the complete dimer for interface evaluation.*

2. The binding pose(s) of oxalate are very intriguing and perhaps the most interesting finding reported in the manuscript. However, both oxalate molecules are built into somewhat ambiguous and weaker densities, and Figure 3C indicates as much. Although the authors perform MD simulations to attempt to validate the binding locations, these are based on the built SLC26A2-oxalate complexes and thus only indirectly validate the binding poses. It can, therefore, not be completely ruled out that these densities correspond to, e.g., other buffer components, though I am inclined to believe the authors. Nonetheless, these unusual but very intriguing binding modes require additional validation, and I am suggesting that the authors perform oxalate transport experiments. Testing different substitutions for S245, S399, or L253 in the vicinity of oxalate should affect oxalate transport or dissociation of transporter-substrate complexes if the proposed binding sites are correct.

Response: *We understand the reviewer's concern regarding the validity of oxalate ($C_2O_4^{2-}$) binding and provide three more analyses to confirm our findings. 1) Map quality enhancement: we reprocessed the SLC26A2- $C_2O_4^{2-}$ data in cryoSPARC v4.4.0 with a newly released feature called referenced-based motion correction, which allowed us to improve the final resolution to $\sim 3\text{\AA}$ (Figure S8 and methods). In addition, using the map improvement tools integrated into the Phenix package, we obtained a cleaner density for the bound $C_2O_4^{2-}$ molecules. 2) Q-score analysis (Lines 202-207): to assess the agreement between the experimental density and modeled $C_2O_4^{2-}$, we calculated the Q-scores in ChimeraX¹. $C_2O_4^{2-}$ -1 (closer to the canonical substrate-binding pocket) scored 0.77, while $C_2O_4^{2-}$ -2 scored 0.53. With the reported resolution of $3\sim 3.5\text{\AA}$, the average Q-score of a model is expected to be ~ 0.6 . Thus, these Q-scores for both $C_2O_4^{2-}$ are acceptable, and their differences align with the observation in MD simulations that $C_2O_4^{2-}$ -1 binds stronger than $C_2O_4^{2-}$ -2. 3) Mutagenesis studies: we conducted mutagenesis studies involving the suggested residues. A triple mutant (S245A/S254A/S299A) exhibited a $\sim 40\%$ reduction in transport efficiency, confirming the significance of these residues in $C_2O_4^{2-}$ transport (Figure S9A).*

3. Chloride, on the other hand, does not fit at all into the provided density, and Figure 3A shows as much. However, there is a clear density for chloride in the vicinity of the current position, and this could be remedied by simply moving the ion. The pocket for this ion is generally not very well built, and key residues such as Q125 need to be better fit into the density. Overall, these issues should be comparatively easy to fix and would much improve the quality of the model. That being said, that does not solve the issue of how chloride is actually coordinated as the authors already state that they do not observe any strong interactions with surrounding residues. Are there potential OH ions involved? Have the authors tested that in MD simulations? How does this compare to Chloride or anion binding sites in other SLC transporters? The chloride must be coordinated somehow, as it is currently just "floating in space."

Response: *1) Map correction: We apologize for the incorrect map previously provided. In this resubmission, the correct map has been included with a better-modeled PDB. 2) Fitting of Cl^- : Contrary to the reviewer's observation in Figure 3A, a snapshot from Coot demonstrates the perfect fitting of Cl^- into the density (Fig a). In addition, the Q-score for Cl^- is 0.83, confirming the model's validity. 3) Cl^- coordination: Cl^- is primarily stabilized*

Fig a. The experimental densities of Cl^- ion and neighboring residues (similar view as the main text Figure 3A)

by the weak dipole introduced by TM3 and TM10, with additional assistance from nearby hydrophilic side chains S396 and S398, as shown in two SLC26A5 structures (PDB: 7LGW and 7V73)²⁻⁴. However, in SLC26A2, these two Ser residues are substituted with Ala (A439 and A441), which likely explains the differing positioning of Cl⁻ compared to SLC26A5. Within the binding pocket, we also observe several much weaker densities, likely representing water molecules. It is plausible that Cl⁻ is stabilized in the pocket with the help of these water molecules rather than being “floating in space.”

4. The authors functionally characterize pathogenic mutations in the “Mapping the pathogenic mutations” section. This section is very confusing. For instance, only selected mutations have been characterized. What is the justification for only testing the mutations shown in Figure S5? The authors also mention a list of mutations in Line 215/216 that they claim likely affect transporter folding. Shouldn't this be tested by ELISA or Western blotting? The authors then say that an R279W mutant has previously been shown to exhibit 50% reduced surface expression and a 50% reduced transport rate but then state that this shows that the mutant 279W does not dramatically affect transport function (Line 250/251). I would consider a 50% reduction in function and trafficking dramatic for sure. But then it gets even more confusing. The authors next test this mutant themselves and state that they also observed “a ~50% reduced transport and ~55% reduced expression level of the R279W mutant compared to the wild-type (Figure S5)” (Line 253/254). First, Figure S5 doesn't even show any expression level data, and 55% is an oddly precise number for any expression level determination. Second, the R279W mutant doesn't even show the claimed 50% reduction in transport, as the transport efficiency seems comparable to wt according to Figure S5. Also, Figure S6 is mentioned in the text before Figure S5.

Response: 1) Only mutations shown in Figure S9 (previously S5) were tested due to sufficient protein expression levels for analysis, unlike others listed which affect protein folding as exemplified in now lines 231. As an example, we tried to express G484D and observed negligible expression in cells (data not shown). 2) The R279W mutant's ~50% reduction in expression level but not in the protein's substrate translocation efficiency aligns with one of the two prior studies; our language has been clarified to reflect this. 3) Transport efficiency data in Figure S9 have been normalized against expression levels. This approach precludes the need to show expression data separately, ensuring a more straightforward interpretation of transport efficiency. 4) We have corrected the sequencing of the Figures.

5. How do the authors arrive at the claim that the “expression level of Δ 1-45 remained at approximately 50% compared to the wild-type”? Figure S3 shows a pretty rough-looking western blot/gel (not clear since not described), and it is unknown how the 50% was calculated. Maybe ELISA would be better to more accurately quantify surface expression levels. Also, there seem to be large aggregation spots on the blot/gel. Is it possible that the expression levels of wt and Δ 1-45 are similar, but the n-terminally truncated construct aggregates during extraction or exposure to detergent in the gel running buffer?

Response: The expression level of Δ 1-45 was determined quantitatively using Image Lab Software from Bio-Rad, which analyzes band intensity. We acknowledge that the initial western blot lacked clarity. To address this, we conducted a repeat experiment, which produced a clearer blot, now presented in Figure S4. This revised blot supports our initial claim more robustly. Additionally, we will consider using ELISA to assess surface expression levels with higher precision in future studies if necessary.

6. The authors analyze cleft sizes in Figure S7 and line 312 forward. Some of these distances, such as G166/A441, are less than 0.4Å. Given the resolution of >3 Å for the structures, there is considerable error associated with atom positions in the models (usually around 0.3-0.5Å for a 3Å structure). The authors need to acknowledge that their calculated differences in binding cleft size have an error associated that, in this case, substantially affects the outcome.

Response: We agree with the reviewer regarding the inherent error in distance measurements. This is now noted in the revised manuscript to ensure the reader understands the calculated binding cleft sizes' precision limits (Lines 342-344). While previous SLC26A5 studies have made similar measurements and subsequent biological claims^{3,4} in the case of SLC26A2, we believe the slight variations in pocket size observed are deemed biologically insignificant. These findings do not suggest a substantial conformational change, a perspective that is now explicitly stated, aligning with the reviewer's observation.

7. The title definitely needs improvement and be focused on a big finding(s) rather than a generic description.

Response: Thank you for the suggestion. In line with the significant findings of our study, we propose the revised title: “Substrate Binding Plasticity Revealed by Cryo-EM Structures of SLC26A2.”

8. In line 114, the authors state: “The two sub-domains undergo an elevator-like rigid-body movement.” I’d like to see this “movement” in figures.

Response: This movement is now shown in Figure S2.

9. In line 118, the authors state: “Due to the lower local resolution, we could not unambiguously identify the specific lipids.” I don’t think the discussion of lipids that are not unambiguously identified adds much to the paper. If the authors choose to include it, they should also show densities in a supplementary figure and discuss the positions and findings from the TLC studies in light of observations made for other SLC26 or related transporters for which lipids have been identified.

Response: We have added a new panel to Figure S3 displaying the lipid densities around SLC26A2. The value of the TLC experiment is two-fold: a) lipids tightly bound to the transporter are real; b) these lipids are a mixture of different types instead of just cholesterol, as implied in studies of other SLC homologs⁴.

10. Figure 2 needs to be strengthened. Comparison of the vertical position of TM3/TM10 in SLC26A2/SLC26A9/SLC26A5/SLC4A1 needs to be expanded to show full protomers. It would also be good to see the differences among the channel pores across the 4 transporters. In addition, the authors need to compare the entrances at both the extracellular and intracellular sides among these 4 transporters to showcase differences and similarities and better highlight an inward-facing state.

Response: Thank you for the valuable suggestions to enhance Figure 2. We have now expanded the representation in Figure S5A to include the full protomers. We have also utilized MOLEonline to calculate and illustrate the substrate-translocation pores for these transporters, which is now presented in Figure S5B. These modifications provide a clearer view of the inward-facing pore in SLC26A2.

11. Figure S1 only shows the processing workflow for one structure. Workflows for all three structures are needed.

Response: Agreed. We have amended the supplementary information to include the processing workflows for all three structures in Figures S1, S7, and S8.

12. In Figure S7, please change the colors of the structures to make them easier to distinguish.

Response: The figure is now S11 and updated with distinct colors: SLC26A2-Cl⁻ in green, SLC26A2-SO₄²⁻ in blue, and SLC26A2-C₂O₄²⁻ in red.

13. Line 317: “Recent studies on SLC25A5 have reported...” Do the authors mean SLC26A5?

Response: Thank you for catching that typographical error. It is corrected now in Line 338.

Reviewer #2’s comments:

1. The SO₄ is of good quality. However, the position of Cl⁻ should be adjusted since it does not align correctly with the density. The presence of C₂O₄²⁻ is likely the result of noise and should be examined carefully. As you can observe, the position of C₂O₄²⁻ is not structurally reasonable (no positive charged residues), and there is also similar noise density in the structure of SLC26A2-SO₄²⁻.

Response: We again apologize for supplying the wrong SLC26A2-Cl⁻ map, which led to the confusion also noted by Reviewer #1 (point 3). Now, the correct map is provided and shows perfect fitting for Cl⁻. As for the C₂O₄²⁻ densities in our structure (also noted in Reviewer #1’s point 2), we have further analyzed the local environment of the C₂O₄²⁻ binding site and found that it is stabilized by interactions with surrounding Ser residues (S245, S254, S399) and backbone carbonyls. The lack of strong positively charged residues at this site is likely intentional because excessive charge-charge interactions could impede the necessary movement of

substrates during translocation. This design is consistent with the need for a delicate balance between substrate binding affinity and transport efficacy.

2. p165-170: The description for Cl⁻ should be updated after the model adjustment. Since the density is not of high quality, the author should consider using “potentially corresponds to Cl⁻.”

Response: With the updated map now included, our reanalysis affirms the accurate representation of Cl⁻ without the need for tentative language. For a detailed explanation of the adjustments and analysis that led us to this conclusion, please see our comprehensive response to Reviewer #1's point 3.

3. p170-174: The MD simulation should be rerun based on the updated model of SLC26A2-Cl⁻.

Response: With the correct map provided, we believe this is not a concern anymore.

4. p185-189: The stabilization of SO₄²⁻ is achieved through the interaction with the positively charged end of the transmembrane (TM) dipoles of both TM3 and TM10. The recognition of SO₄²⁻ by this anion-dipole interaction should be described in the manuscript.

Response: We appreciate the reviewer's suggestion. The description is now added in lines 188-189.

5. p195-205: The densities likely result from noise, and the MD simulation does not support the assignment. If the author cannot provide additional evidence, it is advisable to remove the structure from the main manuscript or relocate it to the supplementary material. Additionally, the discussion should be updated accordingly.

Response: We have thoroughly addressed the concerns regarding the C₂O₄²⁻ densities with comprehensive computational and experimental validations. The details can be found in our response to Reviewer #1's point 2. We believe this body of evidence solidifies the presence of C₂O₄²⁻ in the structure, negating the need for its relocation to supplementary material.

Minor Points:

1) Figure S5: Transport efficiency of SLC26A2 single mutants. No control was provided.

Response: We have updated Figure S9 to include the necessary controls. The transport assay in cells overexpressing wild-type SLC26A2 is a positive control, while the assay in cells without SLC26A2 overexpression is a negative control.

2) I recommend using common names rather than systematic names for SLC26A2 and other SLC proteins. For instance, you can refer to SLC26A2 as "Sulfate transporter" to make the text more reader-friendly and accessible.

Response: We respectfully maintain using the systematic name SLC26A2 in our manuscript. This protein functions as an anion exchanger with specificity for multiple anions, not limited to sulfate. Therefore, referring to it as a "sulfate transporter" does not fully encompass the breadth of its function.

3) For better visualization by readers, the deposited structure should be in dimer format. In cryo-EM, complexes with symmetry are typically built with the entire complex.

Response: We agree with the reviewer's suggestion. The provided structure has been modified to present the protein in its dimer form.

4) Fig 1A should depict the membrane, extracellular side, and cytosol side.

Response: Thanks for the suggestion. We have revised the figure accordingly.

5) Fig1C (p125-130):

Fig1C left panel: The cation- π interaction between R545 and F658 appears to contribute significantly beyond what is shown here. Additionally, the charge-charge interaction between R545 and D660 suggests the need to adjust the direction of the side chain of D660.

Response: We agree with the reviewer's observation. We have updated this figure and the corresponding text (Line 129).

Fig1C middle panel: The figure should include the representation of the main chain for hydrogen bonds; I54 does not form any hydrogen bonds with other residues, although one is depicted here. The side chain of Q is not centered within the density.

Response: *All the hydrogen bonds are calculated in ChimeraX, using relaxed constraints with an angle tolerance of 30° and a length tolerance of 0.6Å. We appreciate the reviewer's suggestion to show the main chain. However, to keep the figure clear and easily interpretable, we have to omit the main chain representation.*

Fig1C Right panel: The orientation of the side chain of S538 should be adjusted in certain structures where it cannot form a hydrogen bond.

Response: *The side chain of S538 has been adjusted in the structures to form a hydrogen bond with the backbone of G534.*

7) Fig2: The position of the reference domain should be included to illustrate the relative motion of TM3 and TM10.

Response: *The B panel serves as a reference for the TM3/TM10 position. In addition, we have added Figure S5 to better illustrate this.*

8) AF models were used in the model building, indicating that it was not a de-novo model building process (p102)

Response: *We have removed the term "de novo" to accurately reflect that AF models were utilized in the model-building process.*

9) “Specifically, residues 53-61 in the N-terminal loop traverse through the shallow groove formed by the two STAS domains, resulting in a buried interface of ~600 Å².” This should refer to the calculation software.

Response: *We have updated the manuscript to cite using ChimeraX for this calculation (Line 124).*

Reviewer #3's comments:

Critical evaluation of the results requires both: (1) more technical details, as well as more robust descriptions of (2) the construction of the systems and (3) their analysis.

(1) what were the simulation parameters used, including temperature; was any additional parameterization of the substrates performed as was done, e.g., sulfate in ref 20; how were the flexible loops missing from the protein structure (Line 104) modeled?

Response: *We have updated the Methods section to include the requested simulation parameters.*

(2) were the substrates initially placed according to the cryo-EM structures, what salt concentration/ions were used, and were individual runs started from the same starting configuration, or were they constructed independently from each other? Showing representative snapshots of the simulation trajectories from the beginning and end can help visualize how the protein is sitting in the membrane relative to the lipids, as well as the relative sizes of the membrane patch and water layer (the provided dimensions of the system don't mean much since the size of the protein structure is not indicated).

Response: *We have incorporated the requested specifics in the Methods section. To better illustrate the simulation setup and results, we have provided snapshots from the initial and final stages of the simulation in Figure. S6A.*

(3) how similar/different were the results from the three replica simulations? For example, what are the errors on the percent time the substrates stayed in their respective binding sites from the independent runs? Was water observed to enter the substrate binding site in all simulations (Line 159)?

Response: *The reported percentage representing the time substrates remained in their binding sites are averages from three independent simulation replicates, with standard deviations now included for each (Lines*

179, 195, 215, and 216). The low standard deviations confirm the replicates' consistency. Additionally, we observed water molecules entering the substrate binding site consistently in all simulations.

- The second category of pathogenic mutations discussed in the text involves residues that “most likely impact protein-lipid interactions” (Lines 238-239). The whole discussion is centered around speculations based on the experimental structure (or what the PPM3 software predicts) but it is surprising that there is no connection to the simulations which provide atomistic resolution into those interactions. Are the known mutations in residues that are in direct contact with lipids in the simulations? How do these contacts evolve during the trajectories, are they dynamic? Showing where the residues are located in a simulation snapshot would be helpful. Can the effect of one or more of these mutations be further confirmed with additional simulations?

Response: Thank you for the insightful comments. Although we did observe lipids interacting with this type of residue, such as W505, in the simulations, we did not detail these interactions for two main reasons. First, our TLC experiment indicated that SLC26A2 interacts with multiple types of lipids. At the same time, our simulations employed a simplified system with only POPC lipids. Given this simplification, drawing conclusions about specific lipid-transporter interactions from our simulation would be speculative and need more experimental proof. Second, this manuscript focuses on elucidating the high-resolution structure of SLC26A2 and its interactions with multiple substrates rather than exploring lipid dynamics. While lipid-transporter interactions are crucial for function, an in-depth analysis would require more extensive simulations beyond the scope of this study and our current capabilities, which could not be completed within a constrained timeframe.

- Along the lines of the specific protein-lipid interactions, do the simulations provide any insights into the ability of lipids to stabilize the interactions between the substrate and the protein? How far are lipid tails/headgroups from any of the resolved binding sites? On Line 302 there is a reference to a “potential translocation pathway”. Can this pathway be mapped onto the structure in relation to the surrounding lipids?

Response: As previously mentioned, this manuscript does not focus on the detailed protein-lipid interactions due to the scope and design of our study. Nevertheless, we have ensured that the updated Figure. S6A illustrates the potential translocation pathway, which does not intersect with any lipid molecules.

- The resolution of the structures is ~3.5 Å, which is above the reported differences between both the locations of the bound chloride vs. sulfate (~2-3Å) and the plasticity of the binding site as measured by changes in pairwise distances in Fig. S7 (~1-2Å). How significant are these differences, then? There is no discussion about the uncertainty in these comparisons, given the structural resolution.

Response: We agree that there are positional errors associated with the model, as in any other protein structures deposited in the PDB. As reviewer #1 pointed out in point 6, the associated error within structures of 3 ~3.5Å resolution is usually 0.3~0.5Å (also reviewed in ⁵). Thus, in our humble opinion, the changes in pairwise distances measured (1~2Å) are insignificant. Please see the details in Lines 342-346.

- Related to the above, the analysis of residues only within 5 Å of the bound substrate in the resolved structures seems very limited. If the chloride ion, for example, were sitting closer to where sulfate binds, that would bring it in contact with some positive residues. What are the residue contacts that the substrates make in the simulations? Analyzing their dynamics would be more informative than looking only at the static structures. Similarly, what are the respective distances from Fig S7 and their uncertainties from the simulation trajectories? Is this plasticity of the binding site observed computationally as well?

Response: In our simulations, detailed in Figure. S6B, we indeed observe a notable flexibility for the Cl⁻ ion within the TM3/TM10 cleft, with its distance to the Ca of G166 fluctuating between 3.5 and 8Å. Notably, at the higher range of this fluctuation (6~8Å), Cl⁻ occupies a position overlapping the SO₄²⁻ binding site. This results in Cl⁻ interacting with residues such as K442, emulating the interaction pattern of SO₄²⁻ (depicted in Figure 3B). This observation suggests that the binding site can accommodate varying Cl⁻ positions, reflecting a level of plasticity that could be functionally relevant. We have incorporated this dynamic analysis into our discussion to provide a more comprehensive understanding of the binding site's behavior (Lines 352-377).

- The statement on Line 291 that in simulations, sulfate is the only one consistently binding within the

TM3/TM10 binding site implies that the ion was initially in solution and then bound to the binding site, which is not what the data in Fig S4 shows. It looks like the ion was initially placed in the binding pocket (as in the structure), and it never came out of it during the 1 μ s simulation. This is different from letting the ion explore different binding sites. The text needs to be edited to remove any ambiguity.

Response: *The reviewer is correct. We have changed the text to “ $S_2O_4^{2-}$ is the only one consistently staying in its original binding site” (Lines 309-310). In addition, the method section is updated accordingly.*

- Similarly, the statement on Lines 303-304 is wrong since no simulations were performed with multiple anions to show that only one binds to the TM3/TM10 cleft.

Response: *There seems to be a misunderstanding. Our simulations of SLC26A2-Cl⁻ demonstrated that, even under conditions with a high Cl⁻ concentration (100 mM KCl) in the system, no more than one Cl⁻ ion was observed binding in the cleft at any time, thereby supporting the statement that simultaneous binding of two Cl⁻ ions is highly unlikely. The corresponding text is now updated (Lines 321-326).*

- Given that SLC26A2 is an exchanger (antiporter), what transport mechanism from known models is consistent with sulfate and chloride binding to the same site in the inward-facing conformation? Is their potentially different affinity for the binding site sufficient to make one come off while the other one binds, or is it possible that there is cooperative behavior as indicated by e.g. the different binding site for oxalate? It is mentioned that different substrates may have different mechanisms of transport but are there such examples in the literature? Discussing these questions would help reconcile the experimentally observed binding modes in the context of the protein function.

Response: *We appreciate the suggestion and added a new paragraph in the Discussion section. Please see more details in Lines 352-377.*

- As stated in the abstract, the molecular mechanism of substrate translocation by SLC26A2 has remained unknown because of the lack of structural information. In at least two places in the text, however, there are statements implying that the mechanism is known and involves an elevator-like rigid-body movement (Lines 114-116 and Line 145). These statements are contradictory and need to be modified/clarified accordingly.

Response: *Thank you for highlighting this point. While it is true that previous research suggests SLC26 transporters commonly utilize an elevator-like rigid-body movement, the exact molecular interactions of SLC26A2 with its substrates have not been fully delineated due to structural data limitations. We have revised the text in the abstract to clarify the distinction (Lines 26-27).*

- In Fig. 2, it is hard to see TM3 and TM10 in the simulation snapshot. It would help if the helices were colored with two distinct colors and the water molecules were shown transparent to more clearly show the binding pocket.

Response: *Thanks for the valuable suggestion. We have updated Figure 2B accordingly. The helices of TM3 and TM10 are now distinctively colored in dark salmon, and water molecules are represented as small red spheres.*

- All plots on Fig. S4 need axis labels.

Response: *We have added the appropriate labels to all plots. This figure is now Figure S6.*

References

- 1 Pintilie, G. *et al.* Measurement of atom resolvability in cryo-EM maps with Q-scores. *Nat Methods* **17**, 328-334 (2020). <https://doi.org/10.1038/s41592-020-0731-1>
- 2 Futamata, H. *et al.* Cryo-EM structures of thermostabilized prestin provide mechanistic insights underlying outer hair cell electromotility. *Nat Commun* **13**, 6208 (2022). <https://doi.org/10.1038/s41467-022-34017-x>
- 3 Bavi, N. *et al.* The conformational cycle of prestin underlies outer-hair cell electromotility. *Nature* **600**, 553-558 (2021). <https://doi.org/10.1038/s41586-021-04152-4>
- 4 Ge, J. *et al.* Molecular mechanism of prestin electromotive signal amplification. *Cell* **184**, 4669-4679 e4613 (2021). <https://doi.org/10.1016/j.cell.2021.07.034>
- 5 Ringe, D. & Petsko, G. A. in *Protein Engineering and Design* (ed Paul R. Carey) 205-229 (Academic Press, 1996).

REVIEWER COMMENTS

Reviewer #1 (Remarks to the Author):

Overall the revised manuscript is much improved from the original submission, and many of my comments have been satisfyingly addressed. That being said, there are still remaining points that need to be addressed:

First, while the S245A/S254A/S299A triple mutant is a great addition to back up the studies of oxalate transport and the resolution of the oxalate structure was improved, there is still considerable ambiguity regarding the oxalate binding poses. The oxalate densities remain weak, but much more importantly, the authors describe two oxalate molecules bound near a surface that not only contains no positively charged residues, but in fact contains a negatively charged aspartate (D250) less than 4 Å away from one of the oxalate molecules. This finding requires an explanation as to how a charge-charge repulsion between D250 and oxalate, as well as between oxalate and oxalate, is resolved in this configuration. Also, the authors suggest that oxalate is transported via a different pathway (line 219-221), but there is a notable absence of discussing why a transporter of divalent anions with a dynamic substrate binding pocket size (Line 332-333) would use a different transport pathway? In light of the ambiguous oxalate densities and the curiously negatively charged surface they are supposedly bound to, this aspect of the presented study remains a weak point.

Second, and as a minor comment, the authors now provide normalized transport efficiency data in Figure S9. While this is useful insight, it needs to be presented alongside both expression and transport data to better illustrate which mutations affect which aspect.

Reviewer #2 (Remarks to the Author):

The author has addressed most of the issues, but one major concern was not properly addressed.

The transport of anions by SLC26A2 is expected to be conserved for Cl⁻, SO₄, and C₂O₄; however, C₂O₄ has not yet been observed to bind to the central pocket. While the author has improved the resolution of the C₂O₄ dataset, the density assigned to C₂O₄ still appears to be incorrect.

Upon closer inspection of the same region in the SO4 dataset, similar density is observed, suggesting that the density attributed to C2O4 is likely due to noise. Consequently, it should not be assigned as C2O4. For structural biology, if the density is not unique to the ligand and does not exhibit reasonable interaction, it cannot be modeled as a substrate.

The two sites of C2O4 may also not be essential for transport, and the use of triple mutants to assess its binding is questionable as three mutations themselves most likely affect the transport process.

If the author relocates this section to the supplementary materials and reduces the description or simply delete this structure, I will be happy the publication of this manuscript.

Reviewer #3 (Remarks to the Author):

The authors have largely addressed my concerns. There are only a few remaining points:

1. The distances shown in the table in Fig. S11 need uncertainties from the replicate simulations. Also, the figure is misnumbered (to S7) in the provided file, which needs to be corrected.
2. Regarding the revised statement on Lines 321-326, while no more than 1 chloride ion is observed to bind in the cleft, the sampling in the simulations is pretty limited since the systems were constructed in a particular way and then run only for 1 us. I would therefore slightly modify the text to: “b) in the SLC26A2-Cl⁻ simulations performed in the presence of 100 mM KCl, we observed only 0 or 1 Cl ions binding in the cleft on the timescale of the trajectories.”
3. Please provide references for the CGenFF server in Methods as indicated on the website:
https://cgenff.silcsbio.com/initguess/summary.php#cite_program

Reviewer #1 (Remarks to the Author):

Overall the revised manuscript is much improved from the original submission, and many of my comments have been satisfyingly addressed. That being said, there are still remaining points that need to be addressed:

First, while the S245A/S254A/S299A triple mutant is a great addition to back up the studies of oxalate transport and the resolution of the oxalate structure was improved, there is still considerable ambiguity regarding the oxalate binding poses. The oxalate densities remain weak, but much more importantly, the authors describe two oxalate molecules bound near a surface that not only contains no positively charged residues, but in fact contains a negatively charged aspartate (D250) less than 4 Å away from one of the oxalate molecules. This finding requires an explanation as to how a charge-charge repulsion between D250 and oxalate, as well as between oxalate and oxalate, is resolved in this configuration. Also, the authors suggest that oxalate is transported via a different pathway (line 219-221), but there is a notable absence of discussing why a transporter of divalent anions with a dynamic substrate binding pocket size (Line 332-333) would use a different transport pathway? In light of the ambiguous oxalate densities and the curiously negatively charged surface they are supposedly bound to, this aspect of the presented study remains a weak point.

Response: We thank the reviewer for the valuable insights regarding the proximity of the second oxalate molecule to D250 and acknowledge the importance of clarifying this point. As noted in our revised manuscript, the electron density for the second oxalate is indeed weaker, reflected by a Q-score of 0.53 compared to 0.77 for the first oxalate. This disparity, alongside molecular dynamics simulations suggesting transient binding of the second oxalate for approximately 30% of the simulation time (significantly less than the first oxalate at 65%), supports the decision to exclude the second oxalate from our final model for clarity.

By omitting the second oxalate, concerns regarding charge-charge repulsion are mitigated, as the shortest observed distance between D250 and the oxalate is approximately 4.5 Å. Furthermore, the overall electrostatic environment closely surrounding the oxalate is relatively neutral, with the exception of the negatively charged D250 (Figure 4E), thus making the binding site feasible. It is plausible that this subtle repulsion from D250 may be instrumental in guiding the oxalate towards the substrate-binding pocket, hence facilitating transport.

In line with Reviewer #2's recommendation, we have reduced the description of this structure and relocated it to a supplementary figure. A short discussion about alternative transport pathways can now be found in Lines 363-369 of this manuscript.

Second, and as a minor comment, the authors now provide normalized transport efficiency data in Figure S9. While this is useful insight, it needs to be presented alongside both expression and transport data to better illustrate which mutations affect which aspect.

Response: Thank you for your constructive comment. To address your suggestion, we have now included the western blots for mutant expression along with the raw transport efficiency data in Figure S10.

Reviewer #2 (Remarks to the Author):

The author has addressed most of the issues, but one major concern was not properly addressed.

The transport of anions by SLC26A2 is expected to be conserved for Cl⁻, SO₄, and C₂O₄; however, C₂O₄ has not yet been observed to bind to the central pocket. While the author has improved the resolution of the C₂O₄ dataset, the density assigned to C₂O₄ still appears to be incorrect.

Upon closer inspection of the same region in the SO₄ dataset, similar density is observed, suggesting that the density attributed to C₂O₄ is likely due to noise. Consequently, it should not be assigned as C₂O₄. For

structural biology, if the density is not unique to the ligand and does not exhibit reasonable interaction, it cannot be modeled as a substrate.

The two sites of C2O4 may also not be essential for transport, and the use of triple mutants to assess its binding is questionable as three mutations themselves most likely affect the transport process.

If the author relocates this section to the supplementary materials and reduces the description or simply delete this structure, I will be happy the publication of this manuscript.

Response: We appreciate the reviewer's detailed analysis regarding the oxalate binding. We agree that the second oxalate is most likely noise. Nevertheless, the presence of the first oxalate density is reasonable, supported by the robust Q-score and its persistence in binding during simulations. In light of this, we have excluded the second oxalate, relocated the structure of the SLC26A2-C₂O₄²⁻ complex to the supplementary figures (Fig. S9), and significantly reduced the accompanying description (Lines 197-213). We trust that these amendments satisfactorily address your concerns and enhance the manuscript's integrity.

Reviewer #3 (Remarks to the Author):

The authors have largely addressed my concerns. There are only a few remaining points:

1. The distances shown in the table in Fig. S11 need uncertainties from the replicate simulations. Also, the figure is misnumbered (to S7) in the provided file, which needs to be corrected.

Response: Thank you for pointing out these issues. We have updated this figure (Fig. S12) as suggested.

2. Regarding the revised statement on Lines 321-326, while no more than 1 chloride ion is observed to bind in the cleft, the sampling in the simulations is pretty limited since the systems were constructed in a particular way and then run only for 1 us. I would therefore slightly modify the text to: “b) in the SLC26A2-Cl- simulations performed in the presence of 100 mM KCl, we observed only 0 or 1 Cl ions binding in the cleft on the timescale of the trajectories.”

Response: We appreciate the reviewer’s suggestion and have changed the wording as recommended (Lines 317-318). Thank you for helping us improve the specificity of our statements.

3. Please provide references for the CGenFF server in Methods as indicated on the website:https://cgenff.silcsbio.com/initguess/summary.php#cite_program

Response: We are sorry for the oversight in our initial submission. As requested (Line 436), we have now included the appropriate references for the CGenFF server.

REVIEWERS' COMMENTS

Reviewer #1 (Remarks to the Author):

Although there remains some ambiguity about the oxalate-bound state of SLC26A2, I find that the authors appropriately framed their findings (Instead, we found weak densities outside the pocket, in the core and gate domains interface between TM5 and TM12, which could accommodate a C2O4²⁻ molecule). Removal of the second oxalate and moving these findings to the supplement also helps with not overstating these tenuous observations. I have no further comments and recommend this manuscript be published.

Reviewer #2 (Remarks to the Author):

The authors have addressed my concerns, and this reviewer has no more questions.